# Analysis of Fowl Adenovirus 4 Transcriptome by De Novo ORF Prediction Based on Corrected Nanopore Full-Length cDNA Sequencing Data

**DOI:** 10.3390/v15020529

**Published:** 2023-02-14

**Authors:** Zhuozhuang Lu, Yongjin Wang, Xiaohui Zou, Tao Hung

**Affiliations:** NHC Key Laboratory of Medical Virology and Viral Diseases, National Institute for Viral Disease Control and Prevention, Chinese Center for Disease Control and Prevention, Beijing 100052, China

**Keywords:** fowl adenovirus 4, transcriptome, full-length cDNA sequencing, RNA-seq, ORF prediction

## Abstract

The transcriptome of fowl adenovirus has not been comprehensively revealed. Here, we attempted to analyze the fowl adenovirus 4 (FAdV-4) transcriptome by deep sequencing. RNA samples were extracted from chicken LMH cells at 12, 18 or 26 h post-FAdV-4 infection, and subjected to Illumina strand-specific RNA-seq or nanopore full-length PCR-cDNA sequencing. After removing the reads of host cells, the data of FAdV-4 nanopore full-length cDNAs (transcripts) were corrected with reads from the Illumina RNA-seq, mapped to the viral genome and then used to predict viral open reading frames (ORFs). Other than 42 known ORFs, 39 novel ORFs were annotated to the FAdV-4 genome. Different from human adenovirus 5, one FAdV-4 ORF was often encoded by several transcripts, and more FAdV-4 ORFs were located on two exons. With these data, 18 major transcription start sites and 15 major transcription termination sites were defined, implying 18 viral promoters and 15 polyadenylation signals. The temporal cascade of viral gene transcription was observed in FAdV-4-infected cells, with six promoters possessing considerable activity in the early phase. Unexpectedly, four promoters, instead of one major late promoter, were engaged in the transcription of the viral genus-common genes on the forward strand. The clarification of the FAdV-4 transcriptome laid a solid foundation for the study of viral gene function, virulence and virus evolution, and it would help construct FAdV-4 as a gene transfer vehicle. The strategy of de novo ORF prediction could be used to parse the transcriptome of other novel adenoviruses.

## 1. Introduction

Adenovirus has an icosahedral non-enveloped virion that packages a linear double-stranded DNA genome of 26–48 kb in length. The family of adenoviridae is classified into six genera, among which Mastadenovirus and Aviadenovirus infect mammals or birds, respectively [1]. Historically, adenovirus was first isolated from in vitro cultured human adenoid tissue in 1953 [2]. To date, human adenoviruses (HAdV), a sub-group of Mastadenovirus, consist of more than 100 types, which were further classified to seven species (HAdV-A to -G) based on the similarity of the genomes. Our knowledge about adenoviruses mainly comes from the study of HAdV-C2 or HAdV-C5.

The life cycle of HAdV in host cells is artificially divided into early and late phases according to the onset of the replication of the viral genome [3]. Virus replication is an accurately programmed process that is regulated, implemented and manifested by the sequential expression of viral genes [4,5]. To compact large genetic information into a relatively small genome, adenovirus organizes viral genes as transcription units, and distributes them on both forward and reverse strands. Generally, a transcription unit comprises a promoter, several viral open reading frames (ORFs), polyadenylation signals (PAS), as well as splicing donors and acceptors. The strategy is typically exhibited in the late transcription unit [4,6,7]. The major late promoter (MLP) of HAdV-5 controls the expression of virus late genes, which encode the proteins involved in virion assembly and all viral structural proteins except pIX. These transcripts, each of which includes an untranslated tripartite leader (TPL) at the 5′ end, are divided into five groups, numbered L1–L5, due to the optional use of five polyadenylation signals. Inside each group, the optional use of splicing acceptors further generates varieties of spliced transcripts; and these transcripts can contain one or more adjacent ORFs, with the leftmost one being translated.

The transcription of HAdV-C2 was studied, and the transcription map was drawn four decades ago [8,9,10]. This map has stood the test of time, and there have been only a few modifications in recent years. In 2007, a new gene, U-exon, was discovered on the reverse strand of HAdV-5 [11]. In 2010, a promoter inside the L4 region was found to control the expression of 22 kD and 33 kD [12]. Recently, three groups combined next-generation sequencing with nanopore-long sequencing technologies to restudy the transcriptome of HAdV-5 or HAdV-2 [4,10,13]. Huge novel low-abundance transcripts were discovered and taken as the background evolution potential for adenoviruses to fit new environments. Price AM et al. even found a fused protein of E4ORF6 and DBP in HAdV-5, which played an important role in the formation of the viral replication center [4].

Fowl adenoviruses (FAdVs), mainly isolated from chickens, belong to the genus Aviadenovirus and currently consist of 12 types (FAdV-1 to -8a and -8b to 11), which are further classified into five species (FAdV-A to -E) [14]. FAdVs are ubiquitous pathogens in chicken flocks and are able to cause diseases, such as adenoviral gizzard erosion (AGE), inclusion of body hepatitis (IBH) and hepatitis-hydropericardium syndrome (HHS) [15]. FAdV-C4 is the major causative agent of HHS, which manifests as an accumulation of clear, straw-colored fluid in the pericardial sac and a swollen, friable, discolored liver with focal necroses and petechial hemorrhages in infected chickens. The first outbreak of FAdV-4 was reported to occur near Karachi, Pakistan, in 1987 [16], and the virus has since spread to most of Asia, Central and South America and European countries. In 2015, the spread of an FAdV-4 variant led to the breakout of HHS in China, which caused significant economic losses to the poultry industry [17,18]. The complete genomes of Chinese FAdV-4 isolates were sequenced, followed by phylogenetic analyses. Compared to the prototypes of FAdV-4 (strains of KR-5 and ON1), Chinese isolates were characterized by a deletion of the ORF19 gene and some point mutations in other genes [17,18]. The mutated fiber 2 and hexon might be responsible for the enhanced virulence [19].

Comparative adenovirology designated adenoviral genes into two classes, genus-common and genus-specific genes, based on the similarity of their protein sequences and evolutionary origins [20]. Genus-common genes, which encode viral structural proteins or those that play direct roles in viral genome replication or virion assembly, are genetically reserved and located centrally in the genome. In contrast, genus-specific genes, which are involved in the virus-host interaction and help create a proper cellular environment for virus replication, are evolutionarily nonconserved between genera and mostly located on both ends of the genome. The viral genes of FAdV-4 have been defined by ORFs and phylogenetic analysis [21,22]. There is no similarity identified in genus-specific genes between Aviadenovirus and Mastadenovirus. Moreover, the genome organization is not totally the same between these two genera. For example, there is an E3 region nested inside viral late genes for HAdV, and such an arrangement is not found in FAdV. However, a systemic study of the transcriptome has not been performed on FAdV-4 or other FAdVs. It remains unknown if the transcription pattern of FAdVs is similar to that of HAdVs.

Combined use of long-read and short-read sequencing technologies makes it possible to elucidate FAdV transcriptome in a relatively short period with reasonable cost [13,23,24,25]. Nanopore sequencing works by monitoring changes to an electrical current as nucleic acids are passed through a protein nanopore, and the resulting signal is decoded to provide a specific DNA or RNA sequence. Its major advantage is its ability to produce ultra-long reads [26]. Nanopore full-length cDNA or direct RNA sequencing are able to identify full-length transcripts and explore isoforms, splicing variants and even fusion transcripts. The inherent weakness of this technique is the high error rate, which can be corrected and ameliorated with the data from Illumina RNA-seq. Here, we attempt to report the transcriptome of a FAdV-4 variant isolated during the outbreak of chicken HHS in China in 2015, after bioinformatic analysis of the corrected cDNA data of the nanopore third-generation sequencing.

## 2. Materials and Methods

### 2.1. Cell Culture and Viruses

Flasks were precoated with 0.1% gelatin (Cat. G9391, Sigma-Aldrich, St. Louis, MO, USA) for seeding LMH cells as suggested by the American Type Culture Collection (ATCC). Chicken LMH cells (Leghorn Male Hepatoma, ATCC CRL-2117) were cultivated in Dulbecco’s modified Eagle’s medium (DMEM) containing 10% fetal bovine serum (FBS; HyClone, Logan, UT, USA) at 37 °C in a humidified atmosphere supplemented with 5% CO_2_ and routinely split twice a week. DMEM containing 2% FBS was used as the maintenance medium for virus-infected cells. The fowl adenovirus type 4 (FAdV-4, NIVD2) was isolated from the liver of a dead chicken during a HHS outbreak at a poultry farm in Shandong province, China. The genome of the isolated virus was extracted and cloned into a plasmid we called pKFAV4 (NPRC2.3.311, https://www.nprc.org.cn/#/Adenovirus/FAdVOne, accessed on 27 December 2022). pKFAV4 was an infectious clone, and wild-type FAdV-4 (GenBank MG547384) was further rescued from a pKFAV4-transfected LMH cell. The rescued viruses were amplified, purified and used in this study [27].

### 2.2. Sample Preparation and Deep Sequencing

Exponentially growing LMH cells were seeded in T25 flasks. When they reached 80–90% confluence the next day, the cells were infected with FAdV-4 at a multiplicity of infection (MOI) of 400 viral particles (vp) per cell for 2 h (the purified FAdV-4 had a particle-to-infectious unit ratio of 200). At 12, 18 or 26 h post infection (calculated from the start of infection), maintenance media were removed by aspiration and cells in each flask were lysed in 1 mL TRIzol reagent (Cat. 15596-018, Invitrogen, Waltham, MA, USA) and preserved at −80 °C. After all samples were collected, the lysed cells were shipped to commercial companies with dry ice for Illumina RNA-seq experiments (Annoroad Gene Technology, Beijing, China) or Nanopore full-length cDNA sequencing (Beijing Dia-Up Biotechnologies, Beijing, China). For the Nanopore full-length transcript sequencing, total RNA was extracted and quantified (one sample for each time point); the RNA integrity was assessed on the Agilent 2100 Bioanalyzer (Agilent, Beijing, China); the library was prepared by using PCR-cDNA Barcoding Kit (Cat. SQK-PCB109) and sequenced on PromethION (Oxford Nanopore Technologies, Oxford, UK); Guppy program was used for basecalling with options of length_100 and quality_7 to generate the raw fastq data; pychopper program was used to identify, orient and trim full-length Nanopore cDNA reads from raw data to generate the clean fastq data; and finally, the raw and clean data were released to the laboratory for further bioinformatic analysis. For Illumina strand-specific RNA-seq, total RNA was extracted from triplicate samples, polyA mRNA was isolated by using the NEBNext Poly(A) mRNA Magnetic Isolation Module (Cat. E7490L, NEB), and the libraries were prepared by using the NEBNext Ultra Directional RNA Library Prep Kit for Illumina (Cat. E7760, NEB). After passing the procedures of quantification and quality control, the libraries were sequenced on the NovaSeq 6000 platform (Illumina, San Diego, CA, USA).

### 2.3. Bioinformatic Analysis of Sequencing Data

RNA-seq data from Illumina sequencing were processed and used as the reference set to correct nanopore reads. Briefly, RNA-seq data were aligned to the FAdV-4 genome (GenBank MG547384) by using Hisat2 with the nounal option to discard unaligned reads [28]. The sam files were converted to bam format with samtools and further changed back to fastq sequences with bedtools bamtofastq function [29,30]. Insilico_read_normalization.pl script in trinityrnaseq package was used to reduce the size of the FAdV-4-aligned Illumina reads [31], and the generated fasta data were used as the reference set for nanopore reads correction.

The host cDNA information was removed from nanopore sequencing data, and the remaining reads were further corrected and mapped to the FAdV-4 genome. In brief, the clean data were mapped to the chicken genome (GenBank GCA_000002315.5) by using the minimap2 aligner [32]. Unmapped reads were extracted by using samtools view with options of -f 4 -F 256 and further converted to fastq format by using bedtools bamtofastq. The normalized Illumina reads served as the reference set, and the host sequence-removed nanopore data were iteratively corrected by LoRDEC with increasing K-mer values as recommended (e.g., k19, k19, k31, k31, k41, k41 and finally k51, k51) [13,33]. The corrected nanopore reads were mapped to the FAdV-4 genome by using the minimap2 program with the options -ax splice -uf -k14 --sam-hit-only.

The sam file was processed with a perl script published previously (classify_transcripts_and_polya_segmented.pl) with arguments of 30 3 -1 no_polya [13]. We chose to use 4 of the 6 generated files: prefix.raw_processed_start_sad_polya.txt was used to predict transcription start site (TSS) and transcription termination site (TTS); prefix.start_sad_stop_pattern_count.txt was a table containing information of mRNA names, TSS, TTS, splicing sites and copy number; prefix.canonical_transcripts_by_abundance.fasta contained the fasta sequences of mRNAs; and prefix.GFF_all_found.gff3 included the gff annotions of mRNAs, which could also be converted into corresponding mRNA sequences in fasta format by using the gffread program [34]. Transcripts were also generated by using stringtie to analyze the sam file with options of -L -R -t [35]. The results of these two programs were similar. Comparatively, the perl script results were neatly organized, and the mRNAs were numbered according to the copy number. Therefore, we used these data generated by the Perl script for the following analysis.

### 2.4. Detection of Core Promoter Elements and Polyadenylation Signals

Major transcription start sites were implied in the Perl script results. A sequence of 201 nucleotides (nt) in length surrounding the TSS (TSS was located at 101-nt) was extracted from the FAdV-4 genome and used to detect core promoter elements by using the ElemeNT program (https://www.juven-gershonlab.org/resources/element/, accessed on 7 December 2022) [36]. Polyadenylation signals (PAS) were searched and labeled 10–30 nt upstream of the transcription termination sites [37,38,39].

### 2.5. Determination of Viral Open Reading Frames (ORFs)

The names of the transcripts, which had a copy number greater than or equal to 5, were extracted from prefix.start_sad_stop_pattern_count.txt. Their fasta sequences were extracted from prefix.canonical_transcripts_by_abundance.fasta by using seqkit grep with options of -n -f [40]. ORFs were extracted from each transcript by using orfipy program with options of –min 150 –max 40,000 –start ATG –strand f [41]. The ORFs for each mRNA were sorted according to the positions of the AUG codon, and the leftmost ORF was treated as the protein encoded by the mRNA. These leftmost ORFs were extracted, the heads of fasta sequences were deleted, and the amino acid sequences were printed with no wrap (one line for each protein) by using seqkit seq with options of -s -w 0. The amino acid sequences were sorted and duplicates were removed by using sort -u command in linux bash, and finally 521 sequences were generated. We compared the previously annotated ORFs (42 ORFs) with the found 521 sequences, and 34 ORFs belonged to both groups. The remaining 487 ORFs, which were different from the known ORFs, were given the numbers F001–F487. The known 42 ORFs were combined. The following analysis implied that 5 more ORFs encoded by low-abundance mRNA might be important, which were also combined to generate a collection of 534 ORF sequences. Novel ORFs were aligned to known ones by using the blat program [42]. Most of these novel ORFs belonged to truncated DBP, hexon, fiber 1, fiber 2 and ORF19A proteins. These ORFs with low copy numbers, short lengths or deeply truncated features were excluded from further analysis. Finally, 105 ORFs were saved for mRNA annotation, and among them, 81 ORFs with a relatively higher copy number or importance were used to annotate the FAdV-4 genome.

### 2.6. FAdV-4 Transcriptome

All mRNA sequences in prefix.canonical_transcripts_by_abundance.fasta were analyzed with orfipy [41], and the leftmost ORFs were extracted and combined, which were aligned to the 105 selected ORFs by using blat. Transcripts encoding identical ORFs were grouped and sorted according to copy number in libreoffice calc (https://www.libreoffice.org/, accessed on 1 December 2022). The total copy number of transcripts was calculated for each selected ORF. The selected ORF name and the total transcript copy number were annotated to the corresponding transcripts in prefix.start_sad_stop_pattern_count.txt, which was further sorted according to strand polarity, TSS, TTS and copy count in libreoffice calc. For each major TSS, the names of transcripts with a high copy number or representative transcripts with diverse TTS were selected, copied and saved. GFF annotations were extracted from file of prefix.GFF_all_found.gff3 with these names, transformed to GTF format by using the gffread program [34], and annotated to the FAdV-4 genome in JBrowse [43,44].

### 2.7. Reverse Transcription Polymerase Chain Reaction and Sanger Sequencing

Total RNA was extracted from FAdV-4-infected LMH at 18 h post infection. RNA of 2 μg was treated with DNase I for 15 min in a volume of 10 μL at room temperature to remove trace DNA contamination (Cat. 18068015, Thermo Fisher Scienfitic, Waltham, MA, USA). DNase I was inactivated by incubating at 65 °C for 10 min after the addition of 1 μL of 25 mM EDTA. Reverse transcription was performed in a reaction volume of 20 μL, including 3 μL of DNaseI-treated RNA, oligo (dT) and PrimeScript reverse transcriptase (Cat: 6110A, Takara, Central African).

The forward and reverse primers were designed to locate in different exons for a transcript according to the parsed FAdV-4 transcriptome. PCR was performed to amplify 7 fragments by using the prepared cDNA as the template with the synthesized primers. PCR products were recovered from an agarose gel after electrophoresis and were sequenced with corresponding PCR primers.

## 3. Results

### 3.1. Statistics on the Nanopore cDNA Sequencing

RNA was extracted from chicken LMH cells 12, 18 and 26 h post wild-type FAdV-4 infection. A nanopore PCR-cDNA protocol was performed to sequence full-length cDNAs (one sample for each time point). After removing the cellular transcripts, the data were corrected with viral short accurate reads from Illumina RNA-seq of the same prepared RNA samples. The schematic diagram of data processing is shown in Appendix A. Statistics on the sequencing data are summarized in Appendix A. The ratios of virus transcripts to total transcripts were 1.1%, 5.8% and 10.3% at 12, 18 and 26 h post-infection, respectively, suggesting a dramatic increase of viral transcription as the culture time prolonged. After nucleotide correction, the reads from three timepoints were combined for FAdV-4 transcriptome analysis.

### 3.2. Transcription Start and Termination Sites Were Inferred from FAdV-4 Full Length cDNA Sequencing Data

After being mapped to the FAdV-4 genome, the positions of full-length cDNAs starting and stopping on the genome were recorded. As shown in Figure 1A, there were some hot spots for the transcription to start. The details of the transcription start were representatively shown at a single nucleotide (nt) resolution in Figure 1B. For example, the copy numbers of transcripts starting from 465-, 466-, 467- and 468-nt were 248, 22,108, 6897 and 800, respectively. The abrupt increase and relatively slow decrease in copy numbers of transcripts before or after 466-nt strongly suggested that this site (466-nt) should be defined as a transcription start site (TSS). Similar situations were seen on other hot spots or on hot spots on the reverse strand (Figure 1B–D). We selected 10 major TSS on the forward strand and 8 major TSS in the reverse direction for further investigation. Generally, they had a record of more than 1000 copies of transcripts that started at single nucleotide sites. These 18 TSS were used to define the corresponding promoters. The transcription termination sites (TTS) were similarly analyzed (Figure 2). Near the TTS downstream region, the copy number of the transcripts dropped even more dramatic when compared to the situation in the TSS upstream region. We focused on 10 major TTS on the top strand and 5 on the bottom for the following analysis.

The surrounding regions of TSS were further analyzed with a promoter prediction program [36]. Transcription initiators were found for all the 18 major TSS; TATA boxes were found in 9 regions, and downstream core promoter elements (DPE) were seen in 7 regions (Figure 3A). TSS of reverse 17,972-nt, 35,495-nt and 43,578-nt seemed to locate in a region containing multiple transcription start sites, where several initiators were found by the software (data not shown). TSS were thus used to designate the corresponding promoters in the following text, e.g., the forward 466-nt promoter. Polyadenylation signals (PAS) were easily found 10–30 nt upstream of the 15 major TTS. Thirteen of them were typical “aataaa”, two were “attaaa” and one was “agtaaa” (Figure 3B) [37,39].

### 3.3. Overview of FAdV-4 Transcriptome and the Selection of ORFs for Genome Annotation

The corrected reads from nanopore full-length cDNA sequencing were mapped to the FAdV-4 genome, and the read coverage was displayed in the coverage track in the integrative genomics viewer (IGV) [45]. The reads from Illumina RNA-seq were similarly mapped and viewed. As shown in Appendix A, the patterns of read coverage with data from different sequencing technologies matched very well, indicating the nanopore data were reliable.

A gene expression cassette can generate transcripts with a slight difference at the 5′ or 3′ ends; a transcribed mRNA may be degraded gradually from both terminals, and those transcripts will be ultimately translated into the same protein. Such facts provide a biological foundation to group transcripts with differences at terminals together and further collapse them into a representative transcript. By observing the copy numbers of transcripts around major TSS and TTS (Figure 1B,D and Figure 2C), it could be found that transcripts located 30 nt beyond the transcription hot start or stopping spots were short in copy number. Therefore, we chose a 30-nt window for TSS or TTS for bundling transcripts, and a previously published perl script was run to classify those full-length viral cDNAs [13]. The generated representative transcripts were sorted according to their copy number and designated a name with a combined ranking position and the copy number, e.g., mRNA#2_29165_copies (mRNA#2_29165cp for short).

The copy number frequency distribution of these transcripts is shown in Appendix A. There were 27,515 transcripts that had just 1 copy, which might mainly represent uncontrolled combinations of exons due to low-frequency noncanonical mRNA splicings. There were 407 transcripts that had a copy number of 5, while there were 2962 transcripts that had a copy number greater than or equal to 5, and the cumulative copy number of these transcripts was 581,091, which represented 93.7% of the total detected viral mRNAs (Appendix A). The canonical ORFs larger than 50 amino acids were extracted, and the leftmost ORF was treated as the protein encoded by each transcript. As shown in Appendix A, all detected transcripts encoded 11,818 proteins, while those with copy numbers greater than or equal to 5 encoded 521 proteins. Because proteins encoded by most low-copy transcripts result from the noncanonical assembly of exons, they should not be the major players in the life cycle of FAdV-4.

We focused on the 521 proteins encoded by transcripts with copy numbers greater than or equal to five. Forty-two ORFs have been annotated in the FAdV-4 genome by bioinformatic analysis before [21], and eight of them were not included in the group of 521 proteins. Five more proteins encoded by low-copy transcripts might also be important. We summarized information of these 534 ORFs (521 + 8 + 5), including protein sequences, mRNA sequences derived from the most abundant transcripts and gff annotations, which were provided in Appendix A.

To select relatively more important ORFs for genome annotation, these 534 ORFs were subjected to further screening. First, newly found ORFs were aligned with the 42 known ones. It was found that most of the novel ORFs were truncated version of known proteins. For example, 56 novel ORFs had similarity to hexon, of which, 30 were truncated hexon fragments. The ORF of F189_39272_cp was a truncated hexon of 382 amino acids (aa) in length, which was saved for genome annotation due to its high transcript copy number. The other 29 truncated hexons were neglected because the transcript copy number was generally 10 times lower than that of the transcript that encoded the complete hexon (31,689 copies). The rest of the 26 ORFs were fusion proteins, with the main part of each protein derived from hexon, and the transcript copy numbers were generally 300 times fewer than those of the transcript that encoded the complete hexon. Such reasons made us take them as a side product of hexon and exclude these hexon-related ORFs from further investigation. Some ORFs had little similarity to known FAdV-4 proteins, but they were also excluded because of their low copy number, short length (less than 70 aa) or nested feature (located inside other ORFs). Finally, 105 ORFs were selected to annotate transcripts, of which 81 were chosen to annotate the FAdV-4 genome due to their relatively higher copy number or importance. The newly found ORFs were named after the neighboring original ORFs. For example, ORF1B-t78aa meant a truncated ORF1B of 78 aa in length, with the “t” in lower case representing a truncated ORF. Similarly, “s” in ORF1-s100aa represented a shift reading frame, “e” in ORF19A-e865aa represented an extended ORF, “h” in 100 kD-h1059aa represented a homologous ORF, while “anti” in DNApol-anti60aa represented an ORF transcribed from the antisense strand of the DNApol gene.

### 3.4. Overall Impression on FAdV-4 mRNA Splicing

The splicing of FAdV mRNA was complicated. The transcripts of genus-specific genes at the left end of the genome on the forward strand were analyzed and exemplified. There were 1119 transcripts that started and terminated in this region. If the TSS and TTS were confined to the major ones (466-, 1057-nt and 1856-, 2739-nt) and the copy numbers were confined to greater than 30 copies, 24 transcripts could be extracted. As shown in Figure 4, besides TSS and TTS, nine splicing sites were found in this region, including four exon start sites (ESS) and five exon termination sites (ETS). Assembly of different exons generated transcript variants, which served as the translation templates for different proteins. Among the selected 24 transcripts, 20 were spliced mRNAs and held a proportion of 84.4% in copy number.

### 3.5. Transcription from Major Promoters

#### 3.5.1. Rules for Transcript Selection and Labeling

The transcription from the 18 major TSS was investigated in detail sequentially. The copy number of a transcript was the first key factor to be considered. To show the main function of each major promoter, the top three mRNAs in abundance for each of the selected 81 ORFs were extracted and displayed in the genome browser if they originated from one of the 18 TSS (Figure 5, Figure 6, Figure 7, Figure 8 and Figure 9). The following two tracks were used to show novel FAdV-4 ORFs newly found in this study and known FAdV-4 ORFs, respectively. More transcripts of these promoters are given in the Appendix A. A transcript was labeled with the following information: its rank in copy number, its copy number, the name of the encoded ORF and the cumulative copy number of all transcripts for the ORF. If a transcript did not encode any one of the selected 105 ORFs, no ORF-related information would be appended.

#### 3.5.2. Transcription Controlled by the Forward 466-nt and 1057-nt Promoters

Representative transcripts, starting from 466-nt are shown in Figure 5 and Appendix A. Nearly all these transcripts included ORF0 on the left. ORF0 contained 240 nt and started at 478-nt on the viral genome. The start codon of ORF0 is located just 12 nt behind the TSS of 466-nt, and some transcripts with a TSS of 478 nt were treated as starting at 466 nt because the grouping policy allowed a collapse window of 30 nt (Figure 1B). ORF0 was small, and it could be an upstream ORF for translation regulation. For all these reasons, the second ORF on the far left in an ORF0 transcript was also labeled. We could see that these transcripts mainly encoded ORF1, ORF1B, ORF1B-t78aa and ORF2 under the condition that the AUG start codon for ORF0 was skipped (Figure 5). It was also seen that the transcription could extend to the very right end of the genome, although the copy numbers of related transcripts were extremely low (Appendix A).

The TSS of 1057-nt is located inside ORF1, which was unexpected for previous genome annotation and resulted in several novel ORFs. These ORFs had a frame shift when compared to previously annotated ORF1, among which ORF1-s86aa and ORF1-s100aa had a considerable copy number and were annotated in the genome (Figure 5). Just as transcripts started at from 466-nt, some transcripts from the 1057-nt promoter also extended to the right end of the genome (Appendix A).

#### 3.5.3. Transcription Controlled by the Forward 9275-nt, 21,891-nt, 25,473-nt and 28,371-nt Promoters

The forward 9275-nt promoter was previously defined as the major late promoter (MLP), which has been predicted and found in FAdV-A1 (CELO virus) in previous publications although the details of related transcripts have not been reported [46]. As shown in Figure 6 and Appendix A, the bipartite leader sequence (BPL) could be accurately defined in most transcripts. The MLP mainly contributed to the expression of 52 kD, pIIIa, penton, pVII, pX, pVI, hexon and protease. These ORFs were contiguous on the genome. The MLP also transcribed considerable amount of mRNAs for fiber1 and fiber2. However, it was relatively less important than the following 2 promoters for other genus-common viral proteins such as 100 kD, pVIII, 22 kD and 33 kD (Figure 6). This was very different from the situation in Mastadenovirus, where the transcription of genus-common genes on the forward strand were mainly driven by MLP. Interestingly, two small ORFs were found in the region between BPL1 and BPL2, which were designated DNApol-anti60aa and DNApol-anti160aa due to their locations on the DNApol region of the reverse strand. However, there was no extremely dominant transcripts responsible for these two ORFs. 956 transcripts encoded DNApol-anti60aa, among which mRNA#938 had the highest copy number of 29. The cumulative copy numbers for DNApol-anti60aa and DNApol-anti160aa were 1608 and 215, respectively, and the former was annotated in the genome. HAdV-C also possesses a small ORF named i-leader (about 150 aa in length) between tripartite leader 2 (TPL2) and TPL3 [9,47]. However, there was no amino acid similarity between these ORFs.

The promoter around 21,891-nt transcribed mRNAs for truncated hexon proteins. The main product would be a truncated hexon of 382 aa in length. It deserved further studying why the virus produced mRNAs in such a great amount for a truncated hexon. It seemed that this was a promoter for short distance transcription since very few transcripts extended beyond the TTS of 23,870-nt although the transcription of mRNA#2_29165cp was remarkably intense. The next 25,473-nt promoter produced mRNAs mainly for 100 kD, 100 kD-h1059aa, pVIII, fiber1, fiber1-t187aa and fiber2. Some mRNAs spliced just 2 nucleotides before the stop codon of 100 kD and generated a novel transcript for 100 kD-h1059aa, which would be translated into a variant of 100 kD and had a difference of 7 aa from 100 kD at the C-terminal (Figure 6 and Appendix A).

The 28,371-nt promoter could be considered as the analogue of the recently found L4 promoter in HAdV-5, which controlled the expression of 22 kD and 33 kD [12]. The ORF of 22 kD had been annotated in FAdV-4 genome before. The transcript of 22 kD could contained an intron, release of which would generate an mRNA encoding a new protein. This protein was larger than 22 kD and was different from it at the C terminal. Compared to the situation in HAdV-5, we named the novel protein 33 kD although the theoretic molecular weight of it was 25.3 kD. The 22 kD and 33 kD in FAdV-4 had similarities in protein sequence to their homologues in HAdV-5, respectively. As in HAdV-5, the transcript of 33 kD of FAdV-4 was much more abundant than that of 22 kD (Figure 6 and Appendix A).

#### 3.5.4. Transcription Controlled by the Forward 35,747-nt, 37,061-nt, 39,979-nt and 42,842-nt Promoters

The promoter near 35,747-nt mainly produced two transcripts. One encoded ORF43, and the other, who contained a short 5′ terminal identical to the former, was called ORF43-h82aa, which was short in coding sequence (CDS). If the leftmost AUG was skipped and the translation started from the second AUG, the transcript for ORF43-h82aa would serve as the template for another viral protein: ORF19A-e865aa (Figure 7 and Appendix A).

The strong activity of the 37,061-nt promoter had been revealed by expressing reporter genes in our previous work [48]. This promoter mainly produced transcripts for GAM1, ORF19A-e865aa and ORF19A-t336aa (Figure 7 and Appendix A). GAM1 had been identified as a functional homologue to human adenovirus E1B19K protein and played a pivotal role in preventing the infected cells from apoptosis in the early phase of the virus life cycle [49,50]. It had been predicted that ORF29 had little possibility to be produced as a real protein [21]. Therefore, mRNA#43 might be a precursor of mRNA#5 and would encode GAM1 after release of the intron. Similarly, the mRNA#48, which contained a short ORF of ORF17-anti70aa at the far left end, could be a byproduct of mRNA#9, which encoded ORF19A-e865aa. The transcripts for known ORF19A were very limited in copy number. In contrast, plentiful transcripts encoded ORF19A-e865aa, which contained 35 extra amino acids at the N-terminal of ORF19A. Just like hexon, ORF19A had a lot of variants. Most of them were shorter than native ORF19A, and those with relatively higher copy numbers were shown (Appendix A).

The last two important promoters on the top strand mainly directed the transcription of ORF19A-e865aa, ORF4-t156aa and ORF4-h171aa (Figure 7 and Appendix A). Two representative mRNAs with difference at the 5′ untranslated region (5′-UTR) were responsible for the translation of ORF19A-e865aa, and the one transcribed by the 39,979-nt promoter contributed more copies. The TSS of 42,842-nt was located 17 nt behind the site of start codon of original ORF4, and this explained why the transcripts for ORF4 were very short in copy number. The main product of 42,842-nt promoter was mRNA#32, which excluded a small intron spanning the stop codon of ORF4 and encoded ORF4-h171aa, a variant of ORF4. Similarly, mRNA#223 could be a precursor or byproduct of mRNA#32 without the excision of the intron.

#### 3.5.5. Transcription Controlled by the Reverse 6405-nt, 8605-nt, 17,972-nt, 25,327-nt and 29,442-nt Promoters

There were 8 major TSS on the reverse strand. The bottom 6405-nt promoter mainly transcribed mRNA#11 for the expression of ORF13-e299aa, which contained 33 more amino acid residues at the N-terminal when compared to original ORF13. mRNA#72, where an intron had not been removed, might be the precursor of mRNA#11. The bottom 8605-nt promoter produced two representative transcripts for IVa2. These two transcripts were different in an intron, which located downstream of the coding sequence of IVa2 (Figure 8 and Appendix A).

Considering the copy number of transcripts, the activity of the reverse 17,972-nt promoter was weak. However, it produced 2 genus-common genes important for the replication of viral genome: DNApol and pTP. The originally annotated pTP protein was 602 aa in length. Only 1 copy of full-length cDNA for it was found. In contrast, relatively more abundant cDNA was found to encode an extended pTP (pTP-e632aa) with 30 more amino acid residues being added to the N-terminal, which might be the major isoform of pTP. The transcript encoding DNApol was also very short in copy number. This promoter also contributed to the transcription of some genus-specific genes such as ORF24-e233aa, ORF13-e297aa and ORF12-e305aa (Figure 8 and Appendix A).

The bottom promoter around the site of 25,327-nt was the strongest one on the reverse strand. Intriguingly, the extremely dominant transcript of it, mRNA#1, encoded a truncated DBP protein (DBP-t440aa). The original DBP contained 493 aa, and the translation of DBP-t440aa used the second methionine codon inside the DBP CDS. The most abundant transcript generated by the next bottom promoter, mRNA#8, also encoded a variant of DBP (DBP-h574aa). DBP-h574aa used the start codon of hypo1 and fused 95 aa from hypo1 to the N-terminal of the protein. Compared to the high copy number of these variants, the transcripts for original DBP were negligible. The bottom 29,442-nt promoter also gave rise to other transcripts of viral ORFs such as ORF24-e216aa, ORF14-e240aa, ORF14-h221aa, ORF14b-e217aa and ORF12-e305aa. These transcripts encoded variants of original genus-specific viral ORFs on the reverse strand and were short in copy number when compared to DBP variants (Figure 8 and Appendix A).

All transcripts coding for DNApol and pTP-e632aa were found and shown in Appendix A. The top 3 transcripts for DBP-h574aa in abundance were also shown. It could be seen that the transcription of DNApol and pTP-e632aa were mainly controlled by the reverse 17,972-nt promoter although sporadic transcripts started around 29,442-nt. The transcription of DBP-h574aa, one major isoform of DBP, mainly started at 29,442-nt, indicating that the transcription of these proteins involved in viral DNA replication was controlled by different promoters in FAdV-4. In contrast, HAdV-5 used 2 neighboring promoters (E2A early and E2A late) to express all E2 proteins including DBP, DNApol and pTP.

#### 3.5.6. Transcription Controlled by the Reverse 35,495-nt, 39,698-nt and 43,578-nt Promoters

The last 3 bottom promoters were relatively weak, especially that of the 35,495-nt. The reverse 35,495-nt promoter functioned locally and directed the transcription of ORF22, ORF20A-e145aa, ORF20 and ORF20-e314aa (Figure 9). In contrast, the other two generated varieties of transcripts, some of which even extended to the far left of the virus genome. The dominant transcripts of 39,698-nt promoter were mRNA#29 and #66, encoding ORF17-t150aa and ORF16, respectively (Figure 9 and Appendix A).

### 3.6. Bird View of the FAdV-4 Transcriptome

In the above subsections, we described the transcription of FAdV-4 genome from the aspect of promoters. Some transcripts, the key mRNAs encoding low-abundance ORFs, were neglected because they were not driven by the 18 major promoters. If we showed the major transcripts from the aspect of the encoded ORFs, it could be helpful to understand the outline of FAdV-4 transcriptome. The most abundant transcripts for the selected 81 ORFs were illustrated in Figure 10. Some conclusions could be drawn from this transcription map. First, most of the transcripts were spliced mRNAs. Second, generally, a promoter controlled the transcription of a group of adjacent genes. Third, although the activity of a promoter could reach a distance very far from the TSS, it became weak there and another promoter would often engage in and improve the transcription of local genes. Fourth, some important virus genes, e.g., pTP and DNApol, were transcribed in low abundance by weak promoters, and some, e.g., ORF24, ORF20A and unk2-141aa, were not discussed in the above promoter subsection. Finally, novel ORF variants were mostly higher in abundance than the corresponding ORFs annotated previously, suggesting more important roles of novel ORF-encoding transcripts in the life cycle of FAdV-4.

### 3.7. Temporal Cascade of Viral Gene Transcription

At the three timepoints of 12, 18 and 26 h post infection, the cumulative copy number of transcripts for a virus ORF was normalized by the total copies of full length cDNA detected (including those from cells and viruses). Sixty one ORFs whose combined unnormalized copy number was greater than 100 were included in the analysis. At 12 h post infection, ORF0, ORF0-related ORFs, ORF19A-e865aa and GAM1 had the highest copies on the forward strand, followed by 33 kD, ORF43, pVIII and fiber2; and DBP-h574aa and DBP-t440aa had the highest transcription on the reverse strand, followed by ORF12-e305aa, ORF13-e299aa, ORF17-t150aa and IVa2. All above mentioned ORFs had a normalized copy number greater than or very close to 100 at this time point. At 18 h post infection, the transcription of most viral ORFs was enhanced except ORF14a-e240aa, ORF14-h221aa and ORF14b-e217aa, and the copy numbers of ORF20 and ORF20-e314aa were very close to those at 12 h. The transcription of protease, fiber2, hexon-t382aa, pVI, hexon, DNApol-anti60aa and ORF19A-t336aa was enhanced more than 20 times. At 26 h post infection, the transcription of 31 virus ORFs kept growing, and normalized copy numbers of 100 kD, 100 kD-h1059aa, fiber1, pVI, hexon and hexon-t382aa increased more than 5 times when compared to those at 18 h (Figure 11A,B). These data indicated that the expression of viral genes was regulated during different phases of the virus life cycle.

The transcription of genes was controlled by promoters. Therefore, we further compared the copy numbers of viral transcripts started from different TSS at various time points post FAdV-4 infection (Appendix A). The major TSS did not change throughout these three timepoints. We noted that the counts of normalized cDNA reads starting from 29,442-nt on the reverse strand were nearly stable (around 1500), suggesting the transcription activity of reverse 29,442-nt promoter stayed constant during the life cycle of FAdV-4 and could serve as an internal control to compare the activity of other promoters in a time-course manner. As shown in Figure 12, at 12 h post infection, the forward 466-nt, 37,061-nt and 39,979-nt promoters as well as the reverse 25,327-nt and 29,442-nt promoters possessed relatively high and comparable activity. If we divided the life cycle of FAdV-4 into early and late phases, the ORFs encoded by the main transcripts of these promoter should be firstly expressed and defined as early genes. At 18 h post infection, transcription activity was broadly improved for all promoters except forward 39,979-nt and reverse 29,442-nt, 35,495-nt ones. The activity of forward 9275-nt, 21,891-nt, 25,473-nt and 28,371-nt promoters increased more dramatically and kept growing till 26 h. These promoters together with the forward 37,061-nt and the reverse 25,327-nt ones possessed relatively higher transcription activity at the late phase.

### 3.8. Confirmation of mRNA Splicing by RT-PCR and Sanger Sequencing

To evaluate the reliability of mRNA splicing events detected in the nanopore full-length cDNA sequencing, 7 pairs of primers were designed and synthesized (Appendix A), and RT-PCR was performed to amplified representative viral mRNAs. As shown in Appendix A, fragments of #3–#7 were properly amplified with expected molecular weights in the first round of PCR. The band of fragment #1 was dim because of dominant amplification of the unspliced isoform. The band of fragment #2 could be contaminated by the product of unspliced isoform. These 2 bands were recovered from gel and used as the template for the second round of PCR (Appendix A). All of the 7 target bands were recovered and subjected into Sanger sequencing. The results were in agreement with that of nanopore cDNA sequencing, and the Sanger sequencing chromatograms for DBP-h574aa and ORF19A-e865aa are shown in Appendix A, respectively.

## 4. Discussion

### 4.1. Nanopore Full-Length cDNA Sequencing was a Reliable Solution for Revealing FAdV-4 Transcritpome when Being Combined with the Accurate Short-Read Sequencing Technology

Oxford nanopore long read RNA sequencing can be carried out with three protocols, including PCR-cDNA, direct cDNA and direct RNA [23], and we chose the procedure of PCR-cDNA in this study. The first step of PCR-cDNA is to convert mRNAs to full-length cDNAs with an oligo(dT)-containing primer to start the reverse transcription and ensure an intact 3′-end of the transcript. After the reverse transcription extends to the 5′-end of the mRNA, several cytidines are added to the terminal of the cDNA in a template-independent manner thanks to the property of the used reverse transcriptase. Strand-switching primer (SSP), which contains several guanosines at the 3′-end, will bind to these cytidines and serve as the succeeding template, and the reverse transcription will continue until the reverse complementary sequence of SSP is incorporated into the cDNA. The template-switching step ensures a complete 5′-end of the mRNA to be included in the cDNA. This template-switching reverse transcription technique is widely used in full-length cDNA cloning (rapid amplification of cDNA ends, RACE) and has been used for more than two decades, indicating its reliability [51,52,53]. In contrast, an inability to read ~15 nucleotides at the 5′ end of each strand and relatively higher error rates were identified as the two principal drawbacks of direct RNA sequencing [24,54]. The PCR step during library building can help increase sequence depth [10], although the quantification accuracy of some transcripts will be compromised because not all genes are equally amplified [55]. The nucleotide errors in nanopore PCR-cDNA data were corrected with parallel Illumina short-read RNA sequencing before FAdV-4 transcriptome parsing.

The distribution patterns also suggested a plausible detection of the major TSS and TTS in the FAdV-4 genome. Theoretically, random degradation or physical truncation at the terminals of mRNAs should generate a symmetrical Gaussian distribution for the position of the 5′- or 3′- ends of a transcript. Conversely, the results of PCR-cDNA sequencing showed an abrupt increase in copy number at a certain position for the 5′-end of transcripts and an abrupt decrease in copy number beyond a certain nucleotide at the 3′-end (Figure 1), suggesting the detection of real TSS and TTS from full-length cDNAs but not terminals of degraded or truncated transcripts. Based on the distribution shape of transcription initiation sites, core promoters are classified into two groups: focused or sharp promoters, which have a single, well-defined TSS and dispersed or broad promoters, which have multiple closely spaced TSSs [56]. We could see that highly active FAdV-4 promoters were mostly sharp promoters, and few were broad ones, e.g., the forward 21891-nt promoter. In addition, promoter prediction supported the finding from PCR-cDNA sequencing data, while polyadenylation signals were also found near all the detected major TTS (Figure 3).

Wet-lab experiments provided evidence to support the PCR-cDNA results. In a previous study, we found two regions in the FAdV-4 genome that possessed promoter activity [48]. By determining the expression of reporter genes, we quantified their strength. The one around the ORF28 had strong rightward activity (half that of the control CMV promoter), the TSS was inferred from the Illumina RNA-seq data, and the transcription factor binding sites were annotated after bioinformatic analysis. The results of PCR-cDNA sequencing in this study were totally consistent with that, and the TSS was located at 37,061-nt. The other one around ORF16-ORF17 region had both rightward and leftward activity, and they should be the forward 39,979-nt and the reverse 39,698-nt promoters. The detected transcription activities were also consistent between these two reports (Figure 12).

### 4.2. A Strategy of de Novo ORF Prediction was Employed to Annotate FAdV-4 Genome and Transcriptome

The known ORFs in the FAdV-4 genome were annotated by bioinformatics analysis and comparative adenovirology [21], and these annotations were mostly not based on or confirmed by experimental data. When we attempted to match the collapsed transcripts to the known ORFs, too much discordance occurred. The fact that most adenoviral transcripts contain more than one ORF made the analysis even more complicated. Finally, we employed the strategy of de novo ORF prediction by recruiting the Orfipy software into the pipeline (Appendix A) [41]. Transcripts with a copy number greater than or equal to five were treated as reliable, true mRNAs; the ORFs encoded by these transcripts were predicted, duplicates were removed and the remaining ORFs were aligned to known ones to generate a concise set of ORFs, which was further challenged by the ORFs predicted from all the transcripts. Some very possible, low-copy ORFs were added to form the final set of ORFs, which was used for transcript grouping and genome annotation. This strategy was different from those used for HAdV-C transcriptome analysis with deep sequencing data because HAdV-C has already been intensively studied and has a reliable transcriptome map for reference [4,10,13]. The strategy of de novo ORF prediction was reasonable and easy to carry out, which could be used to parse transcriptomes of novel adenoviruses in the future.

### 4.3. The Transcriptome of FAdV-4 Was Expanded

The FAdV-4 genome was originally annotated by ORF prediction and comparison [21]. Since it is difficult to predict introns accurately and it has been seen that the known mastadenoviral ORFs are generally located in single exons, these ORFs were predicted to be continuous, uninterrupted sequences in the FAdV-4 genome. However, the results of PCR-cDNA sequencing showed that it was common that ORFs spanned one intron for FAdV-4. Eight and seventeen novel, intron-spanning ORFs were found at the forward and reverse strands, respectively, most of which coded for genus-specific or non-structural proteins. The next group of novel ORFs possessed extended or truncated ORFs of known ones. Generally, these two groups of novel ORFs took up the major proportion of transcripts when compared to the corresponding known ones, emphasizing their important roles in the FAdV-4 life cycle. For example, transcripts for 100 kD-h1059aa, 33 kD, ORF19A-e865aa and DBP-h574aa were 4229, 29,681, 33,484 and 21,922 in copy number, respectively; while the copy numbers for corresponding 100 kD, 22 kD, ORF19A and DBP were 657, 4200, 166 and 75, respectively, which were much lower (Figure 10). Some transcripts encoded novel proteins that resulted from frame shifts of known ORFs due to truncation of the 5′-ends, such as ORF1-s86aa, 52 kD-s105aa and pVIII-s97aa. The last group of novel ORFs did not relate to known ones. DNApol-anti60aa, ORF17-anti70aa, unk1-68aa, Uexon-h205aa and unk2-141aa belonged to this group, and their functions were unknown. Finally, 81 ORFs, including 39 novel ones, were annotated to the FAdV-4 genome.

The expansion of the FAdV-4 transcriptome was manifested not only in increased transcripts for novel ORFs but also in various transcripts coding for an identical ORF. The transcriptome of fowl adenovirus has not been systematically studied before, although the major late promoter of FAdV-C10 was illustrated two decades ago [46]. The FAdV-4 transcriptome was comprehensively revealed in this study, and the results showed that the majority of mature viral transcripts experienced mRNA splicing in FAdV-4 just as in HAdV-5. Diverse transcripts could encode the same viral proteins of HAdV-5. However, this phenomenon was only observed in a few ORFs of HAdV-5. In contrast, it was a prevailing fact for many ORFs of FAdV-4 (Figure 4, Figure 5, Figure 6, Figure 7, Figure 8, Figure 9 and Figure 10). For example, mRNA#3_26851cp encoded hexon, while the cumulative copy number of transcripts of hexon was 31,689 (Figure 10). Both mRNA#7_18229cp and mRNA#15_8247cp coded for fiber2, although they were transcribed from different promoters (Figure 6 and Figure 10).

### 4.4. More Promoters Were Engaged in the Transcription of FAdV-4 Genome

The promoters of HAdV-5 are countable. E1A and E1B promoters control the transcription of the E1 region, neighboring E2 early and late promoters control the expression of E2 proteins, such as pTP, DNApol and DBP, and the transcription of E3 regions is directed by the E3 promoter at the early phase and by MLP at the late stage. The E4 promoter controls the expression of E4 proteins. Two intermediate promoters drive the expression of pIX and IVa2, respectively. Two recently found promoters control the expression of 22 kD/33 kD and U-exon, and the MLP drives the transcription of nearly all genus-common genes on the forward strand [4,10]. In contrast, more promoters were involved in the transcription of the FadV-4 genome, and a different strategy was used by FAdV-4 to control the transcription of genus-common genes.

For FAdV-4, two promoters, 466-nt and 1057-nt, directed the expression of genus-specific genes on the left end of the forward strand. Four promoters, 35,747-nt, 37,061-nt, 39,979-nt and 42,842-nt, controlled the transcription of genus-specific genes on the right end of the forward strand. Promoters of 39,698-nt and 43,578-nt drove the expression of genus-specific genes on the right end of the reverse strand, and promoters of 6405-nt and 8605-nt controlled the expression of a part of genus-specific genes on the left end of the reverse strand and a genus-common gene of IVa2 (by the reverse 8605-nt promoter). The reverse 17,972-nt promoter was the weakest one of the listed major promoters, and it controlled the transcription of the pTP isoform (pTP-e632aa), DNApol and several genus-specific genes (Figure 8). Other important proteins for viral genome replication were DBP variants of DBP-t440aa or DBP-h574aa or both, the transcription of which was controlled by the reverse 25,327-nt or 29,442-nt promoters, respectively (Figure 8). In addition, the reverse 29,442-nt promoter also directed the expression of many genus-specific protein variants on the left end of the reverse strand. Some ORFs, such as unk2-141aa, Uexon-h205aa and Uexon, were controlled by other promoters, which were not described in the results section because of their lower transcription activity.

The transcription of genus-common genes in the core region on the FAdV-4 forward strand was controlled by four promoters, 9275-nt, 21,891-nt, 25,473-nt and 35,747-nt. The MLP (the forward 9275-nt promoter) was the strongest one at the late phase (Figure 12), and its strength could extend to the very right end of the genome to transcribe the genus-specific gene of ORF4 (Figure 6). However, the major influence of the 9275-nt promoter controlled the transcription of adjacent 52 kD, pIIIa, penton, pVII, pX, pVI, hexon and protease and stopped at the protease gene. The main product of the forward 21891-nt promoter was an mRNA coding for an N-terminal truncated hexon protein (hexon-t382aa), which was transcribed at such a high level that it ranked second in abundance among viral transcripts. The forward 25,473-nt promoter controlled the expression of 100 kD, 100 kD-h1059aa, pVIII, fiber1 and fiber2 (Figure 6). Notably, the forward 9275-nt promoter also contributed to a small part of the transcripts for encoding fiber1 and fiber2. The transcription of 22 kD and 33 kD started inside the coding sequence of the 100 kD gene and was under the control of the forward 28371-nt promoter. In HAdV-5, the transcription of 22 kD and 33 kD was controlled by the L4 promoter at the intermediate phase and by the MLP at the late phase [12]. In contrast, the forward 28,371-nt promoter controlled the expression of these two proteins during the whole FAdV-4 life cycle, and the contribution of other promoters was marginal (Figure 6, Figure 10 and Appendix A).

Weak transcription activities were also detected in many other places in the FAdV-4 genome (Figure 1). Their contribution to the transcriptome was small and not discussed.

### 4.5. A Temporal Cascade of Gene Transcription Was Observed in the FAdV-4 Life Cycle

The results from the one-step growth curve showed that FAdV-4 had an average life cycle of 30 h in chicken LMH cells [27]. At the early phase of 12 h, the promoters of forward 466-nt, 37,061-nt, 39,979-nt and reverse 25,327-nt, 29,442-nt had remarkable activity. The key transcripts they produced encoded proteins of ORF0/ORF1, GAM1, ORF19A-e865aa, DBP-t440aa and DBP-h574aa, respectively (Figure 11). At the late phase of 18 or 26 h, all the major promoters except forward 466-nt, 39,979-nt and reverse 29,442-nt, 35,495-nt enhanced their transcription activity. The promoters of forward 28,371-nt and 37,061-nt and reverse 25,327-nt maintained their activity at a high level from 18 to 26 h. The three forward promoters, 9275-nt, 21,891-nt and 25,473-nt, which controlled the transcription of genus-common genes on the forward strand except for 22 kD and 33 kD, even kept increasing their transcription capability from 18 to 26 h (Figure 12).

For HAdV-5, the E1A proteins are expressed at the earliest stage of infection. They work as transcriptional activators and participate in the stimulation of other early promoters of E1B, E2, E3 and E4 and even intermediate L4 promoters [12,57,58]. E1A-like proteins have not been found in fowl adenoviruses. It is possible that FAdV-4 does not have such an E1A-like protein to drive viral promoters, considering the simultaneous activation of forward 466-nt, 37,061-nt, 39,979-nt and reverse 25,327-nt, 29,442-nt promoters at the early phase. It has been revealed that viral IVa2, 22 kD and 33 kD proteins are needed for the full activation of MLP at the late phase of HAdV-5 infection [59,60]. Most likely, the homologs of these three proteins are also involved in the transition from the early to late phases of FAdV-4 infection. In addition, it was implied that there were cis-acting elements for phase transition located in these major FAdV-4 promoters except the forward 466-nt, 39,979-nt and reverse 29,442-nt, 35,495-nt ones (Figure 12).

### 4.6. Flaws of This Transcriptome Analysis

The read depth at the middle genome on the top strand, where the genus-common genes were located, was lower for nanopore data than for Illumina RNA-seq. This phenomenon might result from the heavier bias in the PCR step of amplifying full-length cDNAs, considering that the GC content in this area was higher (Appendix A). Especially, the transcript for compete pIX was not detected because the GC content was as high as 80–90% in some regions inside its coding sequence. Notably, such PCR bias would not seriously influence the analysis of the FAdV-4 transcriptome, although it would make the comparison of transcription intensity between different genes inaccurate.

## 5. Conclusions

Nanopore full-length cDNA sequencing results were corrected with Illumina RNA-seq data and used to analyze the FAdV-4 transcriptome. Eighteen major promoters and 15 major polyadenylation signals were found in the genome. Other than the 42 known ORFs, 39 novel ORFs, most of which spanned introns, were annotated to the genome. Different from HAdV-5, FAdV-4 used four promoters to drive the transcription of genus-common genes on the forward strand, and it used different promoters to direct the expression of DBP and pTP/DNApol on the reverse strand, respectively. In addition, it was very common in FAdV-4 that various transcripts encoded an identical ORF.

## Figures and Tables

**Figure 1 viruses-15-00529-f001:**
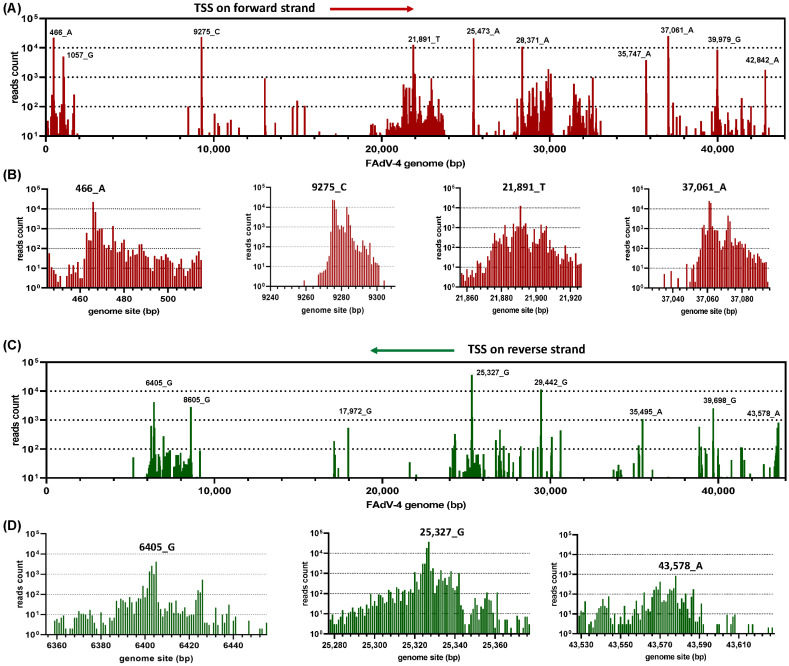
Determination of transcription start sites (TSS) in the FAdV-4 genome. RNA was extracted from FAdV-4-infected chicken LMH cells and subjected to nanopore PCR-cDNA sequencing. Reads originated from the genomes of host cells and were filtered. The remaining full-length cDNA sequences were corrected using Illumina RNA-seq data as the reference and then mapped to the FAdV-4 genome (GenBank MG547384). The resulting sequence alignment/map (SAM) file was processed by a previously published Perl script (classify_transcripts_and_polya_segmented.pl). The start site of each read was collected and expressed as the copy number of reads at each FAdV-4 genome position. (**A**) Read-start sites on the forward strand of the whole virus genome. (**B**) The read-start sites in some representative regions on the forward strand were shown at single-nucleotide resolution. (**C**) Read-start sites on the reverse strand of the whole virus genome. (**D**) The read-start sites in some representative regions on the reverse strand were shown at single-nucleotide resolution.

**Figure 2 viruses-15-00529-f002:**
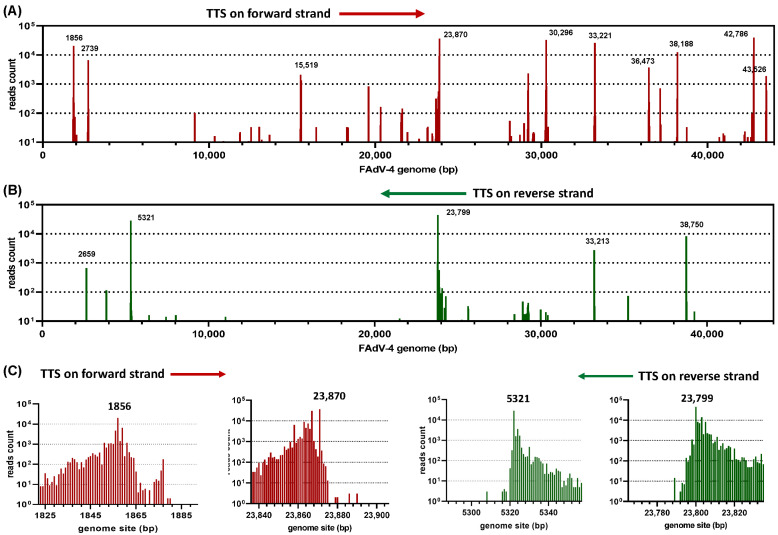
Determination of transcription termination sites (TTS) in the FAdV-4 genome. Viral full-length cDNA reads were aligned to the FAdV-4 genome (GenBank MG547384). The termination site of each read was extracted from the generated SAM file and expressed as the copy number of reads at each FAdV-4 genome position. (**A**) Read-termination sites on the forward strand of the whole virus genome. (**B**) Read-termination sites on the reverse strand of the whole virus genome. (**C**) The read termination sites in some representative regions were shown at single-nucleotide resolution.

**Figure 3 viruses-15-00529-f003:**
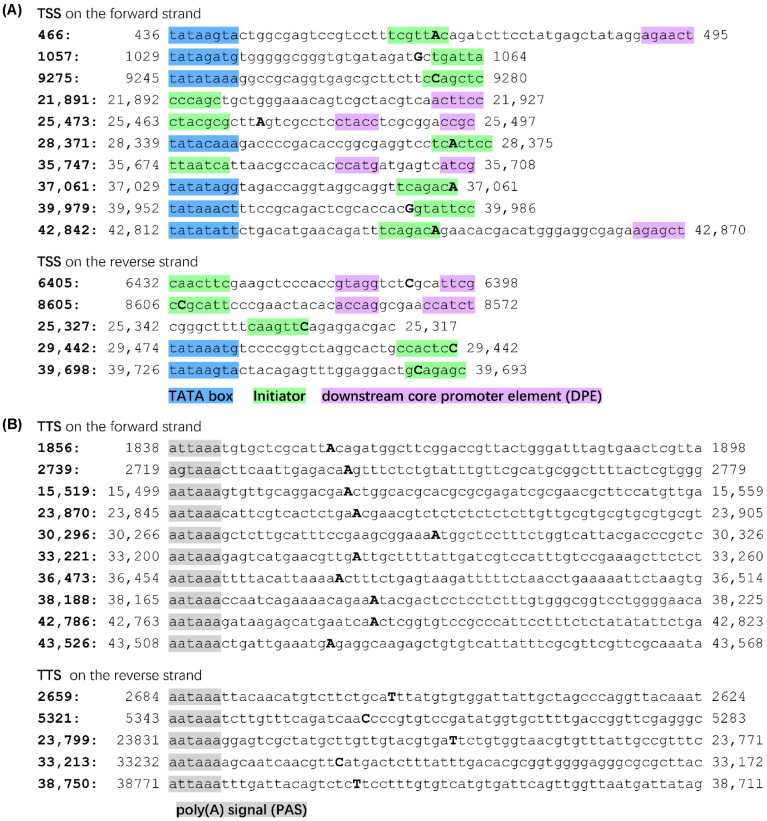
Verification of the found transcription start or termination sites by bioinformatic analysis. (**A**) The sequence around a TSS (201 bp in length) was copied and analyzed by the core promoter element detection program ElemeNT (https://www.juven-gershonlab.org/resources/element/, accessed on 7 December 2022). The elements found were annotated if they helped explain the adjacent TSS. The positions of major TSS in the FAdV-4 genome (GenBank MG547384) were given at the beginning of each line, and the nucleotide at the TSS were shown in bold uppercase in the sequence. (**B**) Polyadenylation signals (PAS) were searched and highlighted 10–30 bp upstream of the major transcription termination sites (TTS) inferred from Nanopore cDNA sequencing. The positions of these TTS were given at the beginning of each line, and the nucleotide at the TTS were shown in bold uppercase in the sequence.

**Figure 4 viruses-15-00529-f004:**
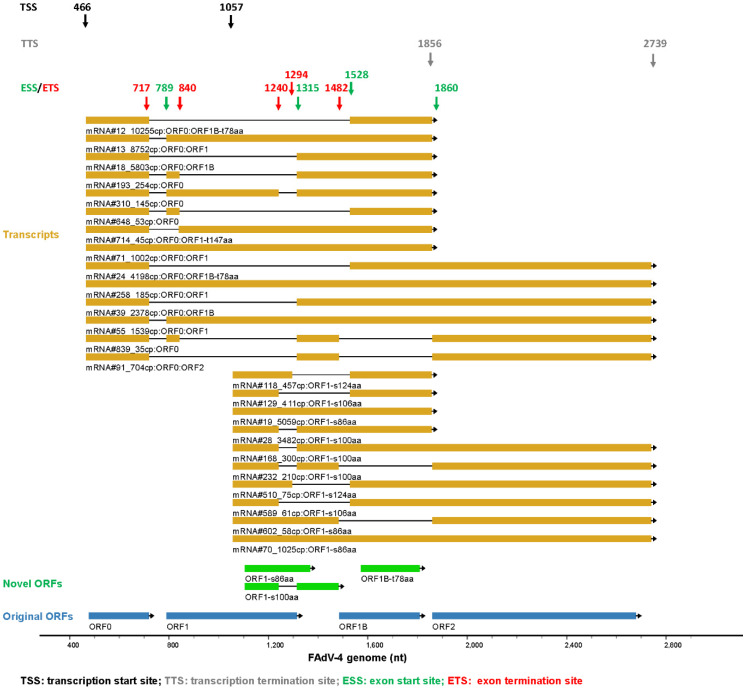
Splicing of mRNAs transcribed from the left end of the forward strand. Transcripts that started at two major TSS (466- or 1057-nt) and terminated at 2 TTS (1856- or 2739-nt) were selected and those with copy numbers greater than 30 were shown. The selective use of splicing donors and receptors generated many spliced transcripts. Since ORF0 could be a small upstream open reading frame (ORF), the ORFs, which started from the second AUG codon at the far-left end of ORF0-containing transcripts, were also labeled. Original ORFs: the ORFs that had been annotated in previous reports. Novel ORFs: the ORFs that were newly found in this study.

**Figure 5 viruses-15-00529-f005:**
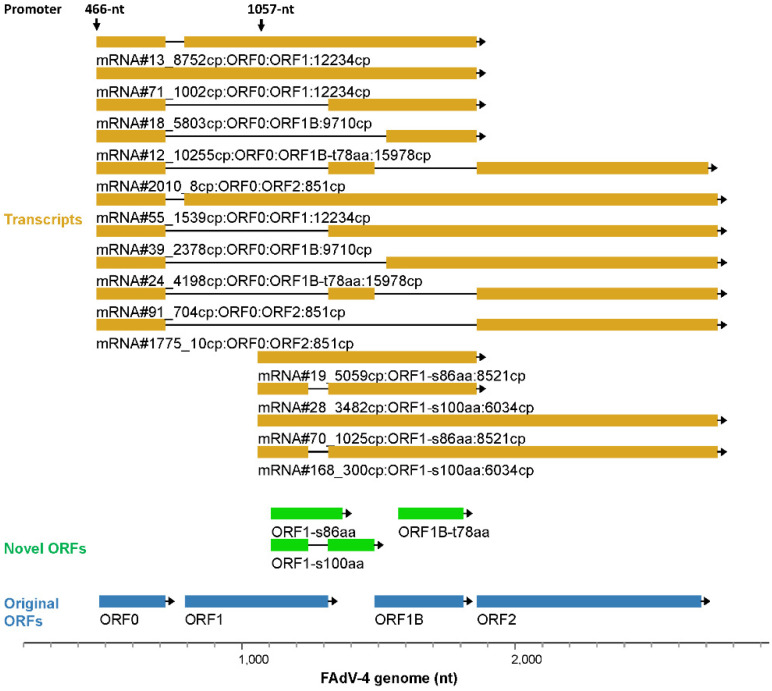
Key transcripts controlled by the forward 466-nt and 1057-nt promoters. A virus ORF was encoded by several transcripts, and the top 3 transcripts of an ORF in copy number were selected. Shown were the selected transcripts that initiated from the major transcription start sites (TSS). The sum of the copy numbers of all transcripts that encoded an ORF was calculated and defined as the copy number of the ORF, which was appended to the end of the ORF name. Original ORFs: the ORFs that had been annotated in previous publications. Novel ORFs: the ORFs that were newly found in this study.

**Figure 6 viruses-15-00529-f006:**
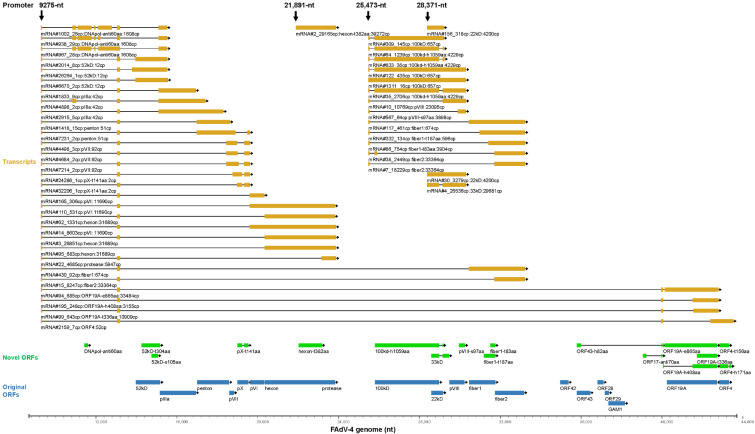
Key transcripts controlled by the forward 9275-nt, 21,891-nt, 25,473-nt and 28,371-nt promoters. The related annotation and explanation can be found in the legend of Figure 5.

**Figure 7 viruses-15-00529-f007:**
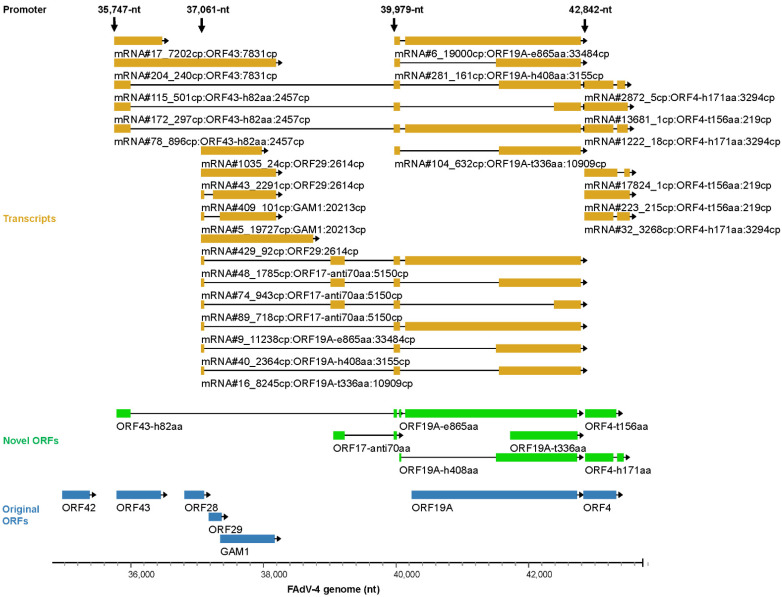
Key transcripts controlled by the forward 35,747-nt, 37,061-nt, 39,979-nt and 42,842-nt promoters. The related annotation and explanation can be found in the legend of Figure 5.

**Figure 8 viruses-15-00529-f008:**
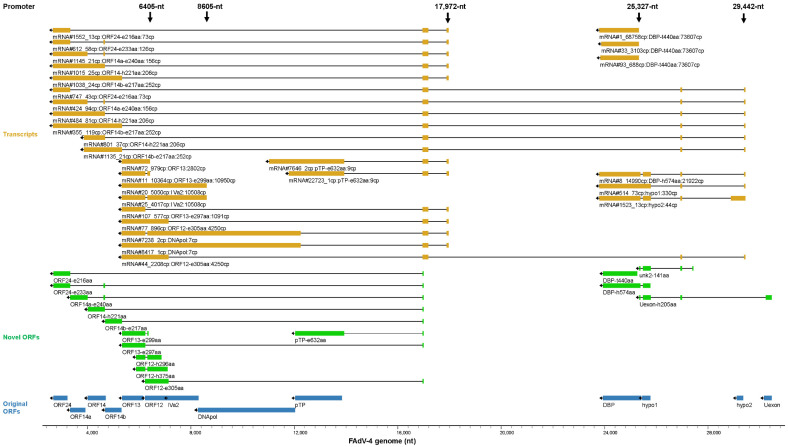
Key transcripts controlled by the reverse 6405-nt, 8605-nt, 17,972-nt, 25,327-nt and 29,442-nt promoters. The related annotation and explanation can be found in the legend of Figure 5.

**Figure 9 viruses-15-00529-f009:**
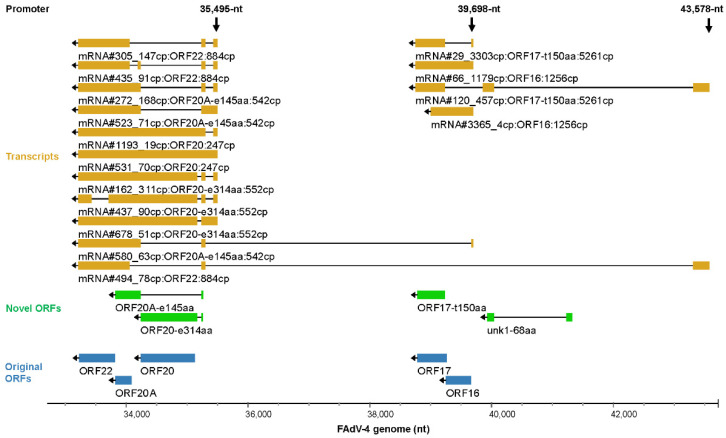
Key transcripts controlled by the reverse 35,495-nt, 39,698-nt and 43,578-nt promoters. The related annotation and explanation can be found in the legend of Figure 5.

**Figure 10 viruses-15-00529-f010:**
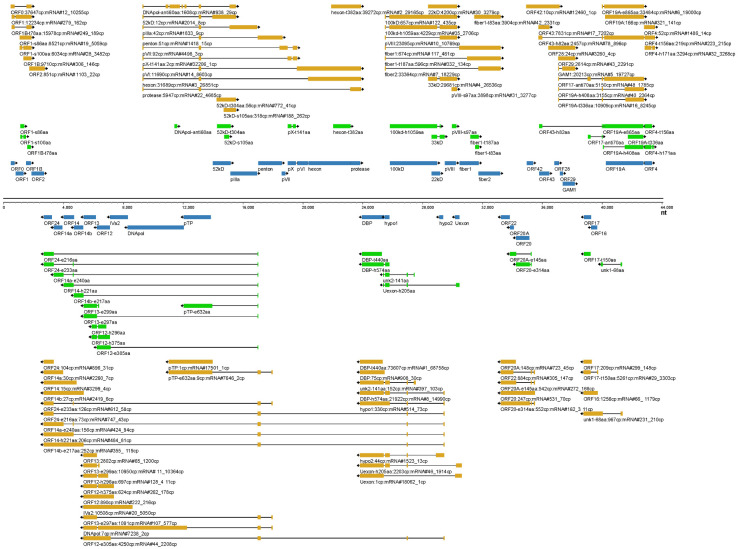
The main transcripts for the 81 selected FAdV-41 ORFs. The most abundant transcripts for each one of the 81 selected FAdV-41 ORFs except pX were annotated. The transcript for pX was not detected because the extremely high GC content in some regions hampered the amplification of full-length cDNA by PCR.

**Figure 11 viruses-15-00529-f011:**
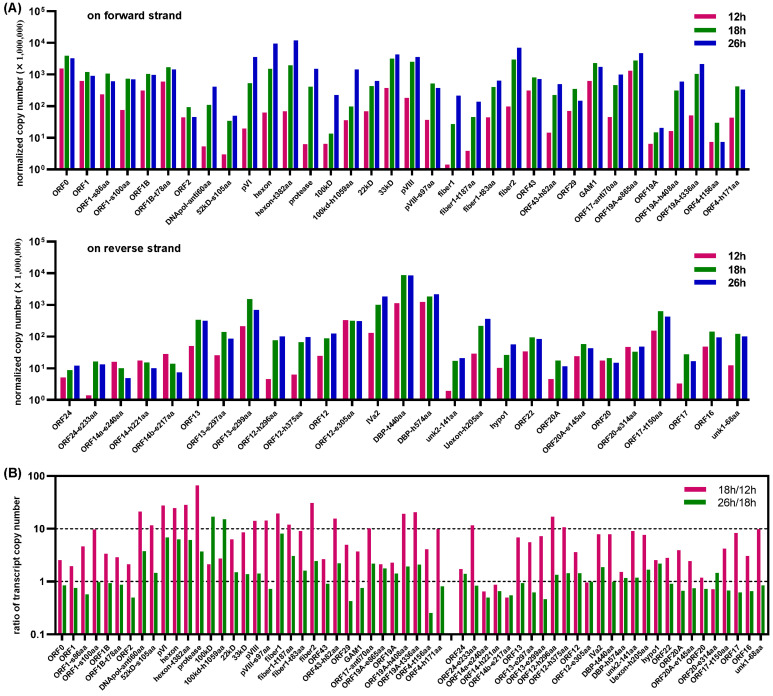
Transcription of FAdV-4 ORFs in LMH cells at 12, 18 and 26 h post virus infection. Virus ORFs with the combined copy number greater than 100 were selected for transcript abundance analysis at different time points post infection. The copy number of each selected viral ORF was normalized by that of total detected transcripts (including those from cells and from viruses, Appendix A) at every time points. (**A**) Transcription of viral ORFs on the forward or the reverse strands at 12, 18 and 26 h post virus infection. (**B**) Change of the transcription intensity for viral ORFs during virus replication.

**Figure 12 viruses-15-00529-f012:**
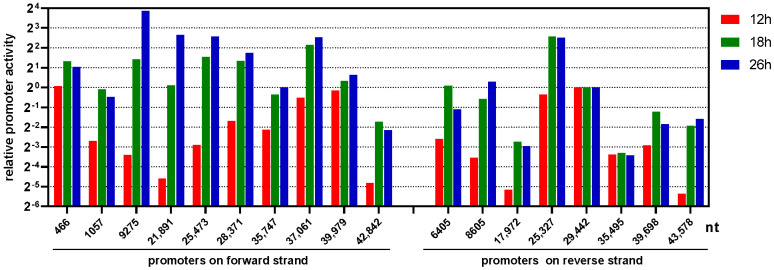
Relative transcriptional activity of promoters. At every time point, the full-length reads, which started within a distance of 30-nt to a major transcription start site (TSS), were counted up and treated as the total mRNAs of the TSS. The total full-length reads at that time point (including those from cells and from viruses, Appendix A) were divided by 1,000,000 and used to normalize the copy number of viral mRNAs. The answers were the normalized reads count, which represented the promoter activity at a TSS. It could be seen that the normalized reads count at 29,442-nt on the reverse strand were 1456, 1502 and 1519 at 12, 18 and 26 h post-infection, respectively, which were relatively stable. The normalized reads count at all major TSS was divided by that at reverse 29,442-nt at various time points, and the answers were relative promoter activity.

## Data Availability

The Nanopore full-length cDNA sequencing data have been corrected, mapped to the FAdV-4 genome, and deposited at NCBI SRA with the BioProject accession number PRJNA805034.

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
