# Peer review of "Analysis of Fowl Adenovirus 4 Transcriptome by De Novo ORF Prediction Based on Corrected Nanopore Full-Length cDNA Sequencing Data"

_viruses, 2023, doi:10.3390/v15020529_

Round 1

Reviewer 1 Report

Lu et al. describe the transcriptome of fowl Ad 4 which was derived from nanopore full-length cDNA sequencing and RNA-seq data. FAdV’s are relevant because of their potential extreme economic impact.  In addition, this work contributes substantially to the description of Avian Ads which have been much less studied on a molecular, genomic, and genetic basis than the mammalian and human Ad viruses.  The authors have done extensive description of the splicing patterns along with copy numbers of the individual transcripts.

I think that figures 1 through 4 are very good and are fundamental to the text.  They express key information for the reader. I did find figures 5 through 15 much more difficult to read and follow.  My sense is that the authors have tried to include too many low copy number transcripts.  This may be to better support their putative novel ORFs, but it makes the figures nearly unreadable.  Somewhat simpler figures would make the paper far more readable (sometimes “Less is more”). Fig 16 seems to be a good overview of the previous figures.  My suggestion is that either the paper needs to be split into 2 papers or that more information should be placed in the supplemental material.  It is good for the authors to confirm many previous ORFs.  It would seem to me that there is too much emphasis on the potential novel ORFs.  At an MOI of 400, it is possible that there are some splicing events that would not be seen in an infection at lower MOI or that defective genomes could also be transcribed.  But this is a good initial evaluation of splicing events.

It was also unclear how the “key transcripts” were selected—some of these are present in very small copy numbers.

Figures 17 and 18 again address more specifics of the time course of use for the promoters and the likely expression of the ORFs.  This again seems like stronger data.

In the text, the authors list the isolated virus they worked with as a “variant” collected from a dead bird.  It would be good to describe whether this is different than the prototype FAdV-4 virus. Is the full sequence of the variant known?

I would suggest using “noncanonical” rather than “uncanonical” in the text.

Reviewer 2 Report

Adenovirus is a very important pathogen to animals and humans. In this study, authors offered us a sequencing data of fowl adenovirus 4. Therefore, at least, the manuscript should contain introduction to fowl adenovirus 4 and the advantages of the nanopore seqencing. Unfortunately, these information are missing. Moreover, the genetic evolution analysis of fowl adenovirus 4 is necessary.

Round 2

Reviewer 2 Report

pass.